# Inhibitory Effects of (−)-Epigallocatechin-3-gallate on Esophageal Cancer

**DOI:** 10.3390/molecules24050954

**Published:** 2019-03-08

**Authors:** Liu-Xiang Wang, Yun-Long Shi, Long-Jie Zhang, Kai-Rong Wang, Li-Ping Xiang, Zhuo-Yu Cai, Jian-Liang Lu, Jian-Hui Ye, Yue-Rong Liang, Xin-Qiang Zheng

**Affiliations:** 1China-US (Henan) Hormel Cancer Institute, No. 127, Dongming Road, Zhengzhou 450008, Henan, China; lxwang@hci-cn.org; 2Tea Research Institute, Zhejiang University, Hangzhou 310058, China; 11516051@zju.edu.cn (Y.-L.S.); 21716160@zju.edu.cn (Z.-Y.C.); jllu@zju.edu.cn (J.-L.L.); jianhuiye@zju.edu.cn (J.-H.Y.); 3Ningbo Huangjinyun Tea Science and Technology Co. Ltd., Yuyao 315412, China; zhanglongjie0701@163.com (L.-J.Z.); wkrtea321hjytea@163.com (K.-R.W.); 4National Tea and Tea Product Quality Supervision and Inspection Center (Guizhou), Zunyi 563100, China; xlping6009@126.com

**Keywords:** tea polyphenols, anticancer, angiogenesis, DNA methylation, metastasis, oxidant stress

## Abstract

There is epidemiological evidence showing that drinking green tea can lower the risk of esophageal cancer (EC). The effect is mainly attributed to tea polyphenols and their most abundant component, (−)-epigallocatechin-3-gallate (EGCG). The possible mechanisms of tumorigenesis inhibition of EGCG include its suppressive effects on cancer cell proliferation, angiogenesis, DNA methylation, metastasis and oxidant stress. EGCG modulates multiple signal transduction and metabolic signaling pathways involving in EC. A synergistic effect was also observed when EGCG was used in combination with other treatment methods.

## 1. Introduction

Esophageal cancer (EC) is a frequently diagnosed cancer of the digestive tract, especially in men, ranking seventh in term of incidence and sixth in mortality rate worldwide. There were about 572,000 new cases and 509,000 deaths around the world in 2018 [1]. China has the highest EC incidence [2]. The prognosis of EC is quite poor, because most patients present with advanced disease. Although great progress has been made in the diagnosis and treatment of cancer, the mortality rate of EC is still very high, and the worldwide average 5-year survival rate of patients with EC is about 15% to 25% [3]. Among EC patients, most of them are diagnosed as esophageal squamous cell carcinoma (ESCC) or esophageal adenocarcinoma (EAC), and these two main subtypes account for more than 95% of all the malignant esophageal tumors [4]. Currently, surgery, chemotherapy, radiotherapy or a combination of modalities are still the main methods used in the treatment of EC. However, these methods have multiple side effects which certain patients are unable to tolerate. Dietary polyphenols were reported to have suppressive effects on the occurrence and development of many cancer types [5], including EC [6]. Development of low toxicity and efficient chemotherapeutics for EC prevention using natural polyphenols are thus urgently required.

Tea is a popular beverage containing high levels of polyphenols, among which green tea is a popular one, particularly in China and Japan. It is made from the leaves of the plant *Camellia sinensis*. Tea leaves are rich in polyphenolic catechins, with 58.0–183.9 mg/g by dry weight [7]. More than ten kinds of catechins have been detected in different types of processed tea, among which (−)-epigallocatechin-3-gallate (EGCG) is the most abundant, accounting for more than 40% of total catechins in fresh tea leaves [8]. In recent years, EGCG has attracted extensive research interest because of its health benefits including anti-inflammation [9], antiviral infection [10], anti-amyloidosis [11], anti-cardiovascular disease [12] and anti-tumorigenesis activity [13]. Although there have been several review papers regarding the association of EGCG or green tea with cancer risk, including breast cancer [14], cervical cancer [15] and bladder cancer [16] through affecting a number of signaling pathways [17,18,19], only one review paper focusing on the effects of tea on EC prevention was published in 2011 [20] based on our literature search of the Embase, PubMed/Medline and Web of Science databases up to early February, 2019. However, many studies have documented the preventive effects of tea or EGCG on EC since 2012 [21,22,23], and the underlying molecular mechanisms of EGCG on EC remain to be elucidated. Reviewing the recent advances in this topic will be help us to further understanding the relationship between tea drinking and EC risk.

In this review, epidemiologic investigations, studies using cell and animal models regarding to molecular and signaling pathways as well as pharmaceutical synergistic effects associated with green tea or EGCG on EC were summarized. The controversial results as well as directions for further research are also discussed. We expect it will facilitate to develop natural products including tea extracts for EC prevention and treatment.

## 2. Literature Search Strategy

Published literatures regarding to tea or EGCG on EC were searched in the Embase, PubMed/Medline and Web of Science databases up to Dec 13th, 2018. The search terms used were: (1) “tea” or “green tea” “EGCG” or “polyphenols” or “catechins”; (2) “esophageal carcinoma “or “esophageal cancer “or “esophageal neoplasms”; (3) these searched keywords were combined using “and” without restrictions. We retrieved all the searched papers and removed the duplicate references using Endnote. Then, we read the titles and abstracts of the remaining papers to exclude those which were not related to the research topic. When we could not decide whether a paper should be included or not based on the title and abstract, we read the full text of the article. The searched papers were screened according to the inclusion and exclusion criteria listed in Table 1. The screened papers were then classified into epidemiological studies, animal and cell studies, clinical trials and inconsistent results, which will be cited in the corresponding subtitles. For the epidemiological studies, each was critically appraised by a quality scale, namely, the Newcastle-Ottawa Scale (NOS) [24], which consists of three variables of quality: Selection (4 points), comparability (2 points), and outcome (3 points), for a total possible score of 9-points (9 representing the highest quality).

The studies to be included in Table 2 should meet the following criteria: (1) It was a case-control or cohort study. (2) It tested the association between EC risk and tea drinking; the odds ratios (ORs) or adjusted OR values and relevant corresponding 95% confidence intervals (CIs) were reported. If the OR value is not available, *p* value was also accepted. (3) The study was about tea individually or tea was definitely included in the study. (4) It was published in English after 2011. (5) The NOS score ≥ 5. However, the other literatures regarding to the epidemiological studies may be cited in the other part of the paper for background information.

## 3. Epidemiological Evidence

Earlier researchers had declared that drinking green tea could suppress esophageal carcinogenesis (Table 2). Early in 1994, a population based case-control study launched in China had found that drinking green tea had a possible inhibitory effect on EC [25]. The relationship between tea drinking and decreased EC risk was also confirmed by several case-control studies [26,27], prospective cohort studies [28,29], and meta-analyses [30,31,32]. It was also shown that women were especially susceptible to the protective effect [25,31,32,33].

However, in epidemiological studies, the impact of lifestyle, differences of hereditary and other confounding factors can lower the confidence level of cancer preventive effect. Daily drinking tea ≥ 3 cups could increase the ESCC incidence rate [34]. In addition, inconsistent results were observed when tea was consumed at different temperatures [33]. It was reported that the EC risk increased because of the mucosal damage caused by large amounts of sipping, rapid eating and consumption of high temperature food [35,36]. In general, drinking a relatively high quantity of tea at low-temperature had a significantly preventive effect against EC, but the risk increased greatly if green tea was consumed at higher temperatures (70–79 °C), OR = 2.21, 95%CI(1.57–5.53); above 80 °C, OR = 4.74, 95%CI (2.6–10.51) [33].

Another two strong interfering factors are cigarette smoking and alcohol drinking, which may hinder the suppressive effect of tea drinking on EC risk. Gao et al. carried out a case-control study in China and reported that the prevention effect was only found in persons who didn’t smoke, among which most were females [25]. Wu et al. also found a significant negative correlation between drinking green tea and the EC risk among the persons who never smoke or drank no alcohol [37]. After adjusting for confounding factors, the preventive effect of drinking green tea on upper digestive tract cancer was obvious [38].

The preventive efficacy of drinking green tea against EC may be neutralized by the strong side effects of alcohol drinking, smoking, or drinking high temperature beverages. Although a majority of data obtained from epidemiological studies and meta-analyses suggested that EGCG could play a positive role in cancer chemoprevention [20], there was also inconsistent evidence of tea drinking on EC. More epidemiological and clinical studies in consideration of confounding factors are needed to prove the anti-cancer efficacy of green tea consumption, for example, the tea infusion temperature when consumed, the amount and frequency of tea drinking, the effective treatment course, and so on. Table 2 lists the epidemiological studies on the association between tea intake and the risk of EC. 

## 4. Animal and Cell Studies

Many studies on cell lines and animal models showed the inhibitory effects of tea EGCG on EC tumorigenesis (Table 3). Potential cancer preventive effects of tea polyphenols or EGCG on EC includes its anti-proliferation, inducing cancer cell apoptosis, inhibiting DNA methylation and angiogenesis, anti-metastasis and relieving oxidant stress. EGCG was observed to regulate multiple signaling pathways related to carcinogenesis. A synergistic effect between green tea components and other clinic treatment methods was also observed. Anticancer effects of green tea and its bioactive constituents on EC in in vitro and in vivo studies are listed in Table 3.

### 4.1. Anti-Proliferation and Inducing Esophageal Cancer Cell Apoptosis

The proliferation and metastasis of EC cells are important factors leading to tumor growth. One of the major characteristics in advanced EC patients is the infinite proliferation of cancer cells. The benefit of inhibiting proliferation of EC cells, first and foremost, is stabilizing the symptoms of the patients and prolonging more treatment time for the doctor.

EGCG could inhibit cancer cell proliferation and induce EC apoptosis [44,45,46,47,53]. It was reported that EGCG inhibited the proliferation and induced EC cells apoptosis (Ec9706 and Eca109) in a time- and dose-dependent manners by modulating the expression of telomerase activity, caspase-3 protein and membrane potential of mitochondrial [45,46] or through p16 gene demethylation [44]. EGCG treatment inhibited the expression levels of anti-apoptotic protein involving Bcl-2 in the NF-κB pathways, and increased the bax and caspase-3, resulting in the cell cycle arrest in G1 phase [46]. It also induced EC cell lines Te-1 and Eca-109 apoptosis [22].

Green tea extracts rich in EGCG inhibited the proliferation of human Barrett’s esophagus cell BE3 and EAC cells (SKGT-4, SEG-1 and BIC-1), arrested the cell cycle in G0-G1 phase through inhibiting cyclin D1 expression, promoting its degradation or dephosphorylation of Rb [48].

### 4.2. Anti-Metastasis

Adhesion, migration, and invasion are the three crucial steps for EC cells metastasis. EGCG treatment (40 μM) or in combination with curcumin, lovastatin can effectively inhibit the invasion of EC cell lines SKGT-4 and TE-8 [17].

The anti-metastasis effects of EGCG are related to its regulation of carcinogenesis factors. VEGF, a key regulating factor for angiogenesis, plays a pivotal role in tumor cell metastasis. Treatment with EGCG down-regulated the expression levels of VEGF in ESCC cell lines (Eca-109 and Te-1), resulting in the inhibition of cell migration [22]. Matrix metalloproteinases (MMP) are a family of proteolytic enzymes which are involved in a variety of physiological and pathological processes [54]. Their main physiological role is to degrade extracellular matrix (ECM) and most protein components in ECM can be degraded by MMPs, which is very important in cancer cell behavior of invasion and metastasis. Treatment to oral squamous cancer cells (OEC-M1) with >5 μM EGCG significantly inhibited the expression of MMP-13. The mRNA of MMP-13 also showed higher expression in ESCC [55], so it is very likely that EGCG could suppress the metastasis of ESCC cell through inhibiting the expression of MMP-13. EGCG also can suppress tumor metastasis through the regulation of related signaling pathways, such as the EGFR signaling pathway [56].

### 4.3. Inhibiting Tumor Angiogenesis

Polyphenols were reported to regulate the functions of various cancer-related signaling molecules. The inhibitory efficacy of EGCG against angiogenesis has been reported in several studies. VEGF is a highly specific growth factor that could increase vascular permeability, extracellular matrix degeneration, proliferation and migration of vascular endothelial cell, and promote angiogenesis. Treatment with EGCG was shown to decrease VEGF expression in ESCC cell lines (Te-1 and Eca-109) and suppress angiogenesis in the patient-derived tumor xenograft (PDX) model of EC. Furthermore, cyclin D1, a key regulator of cyclin-dependent kinase (CDK) and caspase-3, the most important terminal cleavage enzyme in apoptosis, are also regulated by EGCG [22,47].

### 4.4. Inhibiting DNA Methylation

Epigenetic alteration is reversible. More and more studies have been focused on cancer epigenetics alterations in the last few years, including DNA methylation, gene silencing, genomic imprinting and microRNAs [57]. EGCG can suppress the initiation of EC by inhibiting DNA methylation [49]. Actually, DNA hyper-methylation is involved in various key events in cancer progression, for example, regulating the cell cycle, repairing of the DNA damage, and inducing cancer cell apoptosis [58,59,60]. It was reported that DNA methyltransferase (DNMT) could cause methylation of CpG island (the regions of DNA that contain several CpG sites) and further lead to the condensation of chromosome and transcription inhibition [61,62].

DNA demethylation is an important method of cancer treatment, and studies using DNMT inhibitors have further confirmed its possible efficacy. DNMT inhibitor was used to treat cancer cells and mice. EGCG inhibits DNMT and reactivates methylation-silenced genes in cancer cell lines, and thus inhibiting the cancer cell proliferation, inducing cancer cells apoptosis and reducing the tumor volume compared with the control group [63,64,65]. Fang et al. found that EGCG could act as an effective DNA methylation inhibitor. Treating with EGCG (5–50 μM, for 12–144 h) on human EC cells (KYSE 510) was able to suppress DNMT1 activity, leading to concentration- and time-dependent reversal of hyper-methylation and reactivation of several tumor methylation-silenced genes (p16, RARβ, hMLH1 and MGMT) [50]. EGCG also have been confirmed to induce ECa109 cell apoptosis and inhibit cell growth through p16 gene demethylation [44].

### 4.5. Regulating Cell Signaling Pathway and Interacting with Target Proteins

Various cancer-related signaling pathways and molecules have been reported to be regulated by EGCG, such as blocking the cell cycle in G0–G1 phase [66,67] and inducing cancer cell apoptosis [44,46], suppressing the activity of ERK1/2 [17] and mitogen activation protein kinases (MAPKs) [23,68,69], EGFR phosphorylation [56], activator protein-1(AP-1) [70], DNMT [50] and cell transformation [71,72]. In particular, 67-kDa laminin receptor (67LR), a master regulator of many signaling pathways, has been identified as a direct target of EGCG, which was able to be bound by EGCG, thus exerting anticancer activity [73].

The MAPKs, including MAPK/ERK, stress activated protein kinases (SAPK)/JNK and p38, can transduce multiple signaling pathways, leading to a wide variety of cellular responses, such as cell proliferation, differentiation, inflammation and apoptosis [74]. Gao et al. have reported that EGCG could induce EC cell line Eca-109 apoptosis through regulating the expression of JNK and P38 [23].

In human ESCC cell lines (TE-8 and SKGT-4) and EC tissue samples from the patients, the COX-2, phosphorylated ERK1/2 and c-Jun were overexpressed. However, their expression was significantly decreased after treatment with EGCG (40 μM) [17]. A PDX model test also proved that treatment with EGCG (50 μg/kg daily) could decrease expression levels of phosphorylated ERK and COX-2, accompanying with inhibition of tumor formation and growth [17].

Telomerase is very active in many kinds of cancer cells, and its activity is essential for survival of malignant cancer cells, helping them to have infinite division potential. Telomerase activity was downregulated in ESCC cell lines Eca109 and Ec9706 treated with EGCG [45].

EGCG also shows a tumor suppression mechanism through inhibiting the phosphorylation of HER-2/neu and EGFR in ESCC cell line KYSE 150, resulting in the suppression of growth factor receptor [19]. There was an animal study showing that administration of EGCG (4 mg/kg) in *N*-nitrosomethylbenzylamine (NMBA)- induced EC rats decreased the expression of genes COX-2 and cyclin D1, leading to the reduction of the PGE-2 production [47], indicating that EGCG may exert anti-tumor effects through regulating COX-2 and cyclin D1. Similar results were further confirmed by another NMBA-induced F344 rat model test [75]. These experiments indicated that EGCG may have a preventive effect on EC, and the related mechanism may be through downregulating the inflammatory-related factors, such as COX-2, ERK1/2, and c-Jun. Figure 1 illustrates possible mechanisms of EGCG on EC based on the published literature.

### 4.6. Antioxidant and Pro-Oxidation

In normal cells, EGCG is a natural antioxidant agent, possessing strong scavenging capacity for free radicals including hydroxyl radical, superoxide anion and hydrogen peroxide. One of the prime explanations is that the poly-hydroxyl structure plays an important role [76,77,78]. Tea polyphenols were found to inhibit the production of ROS like nitrogen dioxide (NO_2_), nitric oxide (NO), and peroxynitrite [77]. EGCG, a preponderant constituent of tea polyphenols, could decrease lipid peroxidation and several pro-inflammatory cytokines, and relieve oxidative DNA damage through regulating Nrf2-Keap1 signaling pathway [79,80]. It also increased the generation of some detoxification enzymes, like GST (glutathione *S*-transferase), COX-2, CAT, GPX, and inhibited the accumulation of transcription factors which is redox sensitive in nucleus such as NF-kB, AP-1 in different experimental models [77,81,82,83]. It was found that green tea catechins have a stronger protective effect against the progression of gastric and esophageal cancers among the participants who are lack of serum carotenes [84].

Low concentration EGCG itself can be oxidized to produce ROS and exhibit pro-oxidant and apoptosis-inducing properties in cancer cells because they were unstable [19,85]. For example, it can induce ROS production in ESCC cell lines (Te-1 and Eca-109) [22]. Pretreatment of ESCC cell line KYSE 150 with EGCG before the addition of EGF led to 32–85% reduction of phosphorylated EGFR and 80% decrease in EGFR protein expression. The addition of SOD to the cell culture medium can inhibit or reduce these effects induced by EGCG. EGCG was easy to form dimers or other oxidative products when co-cultured with ESCC cell line KYSE 150. In the culture medium, EGCG was stabilized by SOD, and the ability of EGCG to inhibit cell growth was also enhanced. These studies indicate that the auto-oxidation of EGCG under cell culture conditions may result in the inactivation of EGFR-related signaling pathway [19].

### 4.7. Animal Model Tests

As early as 1992, Chen reported that oral administration of green tea to the rats significantly reduced the esophageal tumors occurrence induced by NMBA [49]. Another experiment confirmed that drinking water containing 1200 ppm EGCG for 14 days before administration of NMBA significantly reduced the rates of esophageal tumor formation [51]. The mechanism is considered to suppress the expression of cyclin D1 and COX-2, and induced the production of PGE-2 [47].

In a tumor mouse model transplanted with Eca-109 cell of ESCC, treatment with EGCG significantly reduced the tumor volume through decreasing expression levels of VEGF protein and increasing cleaved-caspase-3 [22]. Another study found the similar inhibitory effects in a PDX nude mouse model through downregulating the expression of COX-2, pERK1/2, c-Jun and upregulating the expression of activated caspase 3 [17]

Wang et al. reported that administration of 0.6% green tea regularly during or after treatment by NMBA could significantly reduce esophageal tumorigenesis in rats (a reduction of approximately 70% and 50%, respectively), the possible mechanism may be through inhibition of tumor incidence and multiplicity, the tumor volume of the mice was also significantly reduced by regular drinking 0.9% green tea [52]. However, Li et al. reported that drinking beverages at high temperature can lead to increased esophageal carcinoma risk, and thus the anticancer efficacy of EGCG on NMBA induced rat esophageal tumorigenesis may be offset [75].

### 4.8. Pharmaceutical Synergistic Effect

Radiotherapy and chemotherapy are traditionally or frequently used cancer therapy methods, but adverse effects of these treatment methods are hard to bear before the patients have been cured. Therefore, natural extracts with synergistic effect can be used to lower the adverse-effects of traditional cancer treatment methods are urgently needed.

A combination of EGCG, curcumin and lovastatin can significantly enhance the anticancer efficacy compared to using any one of them alone. They inhibited the proliferation and invasion of EC cells (SKGT-4 and TE-8), and reduced the expression of c-Jun, pERK1/2, Ki67, and COX-2, but activated caspase-3 in xenograft tissues [17]. EGCG also inhibited the ABCG2 expression in a multidrug resistance ESCC cell line Eca109. Compared with adriamycin (ADM) treatment alone, the rate of apoptosis of Eca109/ABCG2 cells treated by combination of EGCG with ADM for 24 h enhanced ADM concentration in the cancer cells, suggesting that EGCG can reverse multi-drug resistance by reducing the expression of ABCG2 and increase the concentration of drug in cancer cells to enhance the anticancer effects [46]. Gao et al. confirmed that vitamin C could significantly enhance the therapeutic properties of EGCG through activating the activity of caspase-3/9, inducing cancer cell apoptosis by regulating MAPK pathways [23]. Table 4 lists the pharmaceutical synergistic effects of EGCG with other treatments on EC.

## 5. Clinical Trials

Barrett’s esophagus (BE) patients have a higher EC risk, and currently there are no available methods for cancer prevention in this population [86]. Therefore, such patients must undergo invasive endoscopy, including multiple biopsies. Joe et al. completed a prospective, phase I, placebo-controlled trial in patients with BE using a formula “Poly E” containing green tea EGCG. It showed that EGCG was significantly accumulated in the esophageal mucosa of the target organ after administration poly E (400 mg or 600 mg), which was clinically relevant and detectable, indicating that poly E has a measurable positive protective effect in esophageal tissue [87]. Acute radioactive esophagitis (ARIE) is one of the most frequent adverse effects of chest radiotherapy in the treatment of lung cancer complications, which seriously affects the life quality of patients and the effect of radiotherapy. So far it has been difficult to be overcome quickly. A prospective phase II clinical trial in China confirmed that oral administration of EGCG was an effective therapy method for ARIE, which can significantly reduce the pain of patients and improve the therapeutic effect [88].

One of the major challenges of EGCG for cancer prevention is to discover new biomarkers and use them in clinical practice. Therefore, it is important to identify more targets and biomarkers of tea polyphenols in EC for the implementation of green tea experimental design, and this will greatly help to better understand the mechanism of its activity of cancer prevention.

## 6. Discussion

### 6.1. Inconsistent Results

Although cell and animal studies have shown that tea polyphenols have a positively protective effect on the prevention and treatment of EC, there are also inconsistent results from human epidemiological studies. Many factors related to the consumption of green tea may lead to these inconsistent result, for instance, the type and quantity of tea intake, the temperature of the tea beverage, the low bioavailability of tea polyphenols, and the different etiology of cancer in different regions. For example, a European prospective study showed no significant association between tea intake and risk of EC, EAC and ESCC [40].

In the epidemiological studies, the quantity of tea drinking was mainly investigated, while the temperature of tea infusion when consumed was rarely concerned. Consuming tea at higher temperature might confound the relationship between EC risk and green tea drinking. Yang et al. reported that very daily hot tea drinking significantly increased the risk of ESCC among Chinese men, which is particularly evident among alcohol drinkers [89]. The most recent meta-analysis also showed an approximately 2-fold increased risk for ESCC in association with consumption of hot food or drinks [42]. It has been verified that it was the temperature effect but not tea constituent may pose an EC risk through comparing the margin of exposure of polycyclic aromatic hydrocarbons (PAH) vs. very hot temperatures [90]. Smoking and alcohol drinking might be associated with the inconsistent results. Yu et al. showed that persons who drank hot tea in combination with alcohol and/or smoking had 5-fold times of EC risk [43].

Different kinds of tea have different inhibitory effects on EC. Green tea instead of black tea is generally associated with a reduced risk of EC [32,39,40]. The reason may be that the main biological active ingredient possessing anti-cancer property in tea is catechins, especially EGCG, and most of the catechins are oxidized during black tea processing, during which most of tea catechins are oxidized and condensed into theaflavins and thearubigins.

Identifying specific, objective biomarkers for metabolism of EGCG in human body may help the assessment of relationship between tea drinking and EC risk. Rasool et al. found a significant correlation between the ECRG1 genetic polymorphism and ESCC in Kashmiri population [91]. More different population-based studies are needed to prove the function of the biomarker.

Bioavailability of tea catechins during human tea consumption was very low, and the dosage of daily tea consumption was also lower than those used in laboratory studies. These may be partially responsible for the inconsistent results. Meng et al. reported that oral administration 2 mg/kg EGCG, the maximum plasma concentration was only 0.09 μM, indicating a rather low bioavailability of EGCG [92].

### 6.2. Further Study Directions

The inhibitory effect of EGCG alone on EC is limited. However, combination of EGCG with radiotherapy, chemotherapy, molecular targeted therapy, immunotherapy or other natural compounds has been shown to have a better clinical efficacy in treatment of cancer patients, including EC [93,94]. Green tea has been recognized as an effective sensitizer that can enhance the radiotherapy effect of prostate cancer [95] and breast cancer [96]. However, few researches have been performed on the adjunctive effect of green tea on radiotherapy for EC.

Further study on the synergistic anti-tumor effect of catechins with other bioactive compound and the method for stabilizing catechins in the digestive tract will be helpful to improve the oral bioavailability and anti-cancer effect of catechins. It has been reported that EGCG acetylation can significantly improve the bioavailability in vivo and in vitro of colorectal cancer and EC [97,98].

A meta-analysis reported that there was an increased ESCC risk in the population infected by HPV-16 [99]. EGCG could inhibit the growth and induce cancer cells apoptosis through regulating the expression of HPV oncoproteins [15], so it may be used as an effective dietary agent to prevent infection of HPV by targeting the oncoproteins.

MicroRNA is a class of non-coding, short RNA, which are responsible for regulating the expression of post-transcriptional gene. Several studies have revealed that EGCG had the potential to regulate the expression of microRNA in melanoma [100], colorectal cancer [101] and lung adenocarcinoma [102]. Indeed, there are some dysregulated miRNAs in ESCC [103]. Hence, targeting the microRNAs using tea catechins may be a promising project in EC therapy.

The cancer stem cell (CSC) is a member of cancer cells that have stem cell properties, which plays important roles in tumorigenesis, tumor recurrence, cancer metastasis and chemotherapy resistance. EGCG can target CSCs for cancer prevention and treatment [104]. For example, EGCG treatment significantly inhibited spheroid formation, ALDH activity, and expression of specific genes in human neuroblastoma CSCs, such as Oct 4 and Nanog [105] and the tumor volume in PDX models [106]. Chen et al. discovered that EGCG could suppress the spheroid formation of colorectal CSCs by inhibition of the Wnt pathway [107]. Recently, Kumazoe et al. reported that combination treatment of EGCG with phosphodiesterase 3 inhibitor successfully suppressed CSC properties in pancreatic adenocarcinoma [108] and Chani et al. found that EGCG significantly inhibited mesenchymal stem cell differentiation to adipogenic lineage [109], suggesting that EGCG may intervene in either obesity or metabolic syndrome-associated cancer. Till nowadays, there have been very rare studies on the effect of EGCG on CSCs of EC, therefore, targeting CSCs with EGCG for EC prevention and treatment may be a promising project in future.

P53 is the most common mutation gene in ESCC, and about half of EC patients have P53 gene mutation or overexpression, even in the early stage of cancer [110,111]. It is considered as an important biomarker in the diagnosis and therapy of ESCC [112,113]. P53 was a good predictor for therapy response and survival, and the overall 5-year survival rate of the ESCC patients was significantly higher in the p53-normal group than the p53-alteration group [113,114]. Thus, P53 is a promising target for anticancer drugs. EGCG also could suppress the growth of breast cancer cells and induce apoptosis through regulating P53/Bcl-2 signaling pathway [115]. However, the inhibition mechanism of EGCG for EC needs more research to support the efficacy in the future.

Compared with normal cells, the cancer cells exhibit some different biophysical properties, such as cell stiffness and elasticity, and the cell stiffness has been identified as a new biomarker of the metastatic potential in cancer cells [116]. Recently, Suganuma et al. have made some new discoveries in cancer research by using biophysical tools, such as atomic force microscope (AFM). In their study, they found that after treatment with EGCG, the melanomas and various cancer cells have a significantly lower average value of Young’s moduli, indicating that EGCG may inhibit cell migration and metastasis through increasing the stiffness of cancer cells [117]. We suggest that more study on the biophysical properties of cancer cells are needed because this will play an important role in early diagnosis and prevention of cancer cell metastasis.

Drinking one cup of green tea (containing about 200 mg EGCG) daily has been shown to have chemo-preventive effects on several kinds of cancer [118]. The proper drinking amount of green tea is advised to have three to five cups per day, which is equivalent to about 600–1000 mg EGCG [119]. Suganuma et al. proposed that drinking over 10 cups of green tea per day (equivalent to 25 g green tea) can prevent the recurrence of various cancers [117]. Based on the average content of EGCG in dried green tea (about 80 mg/g) [120], the results of epidemiological and clinical trials, and in consideration of the extraction rate of the tea catechins, we recommend that daily drinking 5–7 cups of green tea (15–20 g) at a lower temperature may be a positive method for the prevention of EC. More clinical trials and epidemiological studies considering the drinking amount are needed to support the preventive effects of EGCG on EC. Furthermore, it was reported that tea catechins inhibited influenza viral adsorption and suppressed replication and neuraminidase activity. Tea catechins enhance immunity against viral infection [10], suggesting that the cancer inhibition effect of tea catechins might also be indirectly linked with cancer progression as catechins have good influence on human health.

## 7. Conclusions

In this review, we summarized the advances in the possible antitumor effects of green tea polyphenols or EGCG in EC prevention and treatment. Possible mechanisms of EGCG against EC included inhibition of EC cell proliferation, DNA methylation, angiogenesis, enhancement of apoptosis, anti-metastasis, antioxidant or pro-oxidant, synergistic effect in combination with other treatment methods as well as the modulation of multiple signal transduction and metabolic signaling pathways involved in carcinogenesis.

Some epidemiological studies showed inconsistent results, these inconsistencies can be attributed to differences in temperature of tea solution when drinking, the quantity of green tea intake, and the influence of other factors, such as smoking, alcohol drinking. More studies are needed to identify the precise preventive mechanisms of EGCG on EC. Overall, EGCG shows an inhibitory effect on EC in laboratory studies.

## Figures and Tables

**Figure 1 molecules-24-00954-f001:**
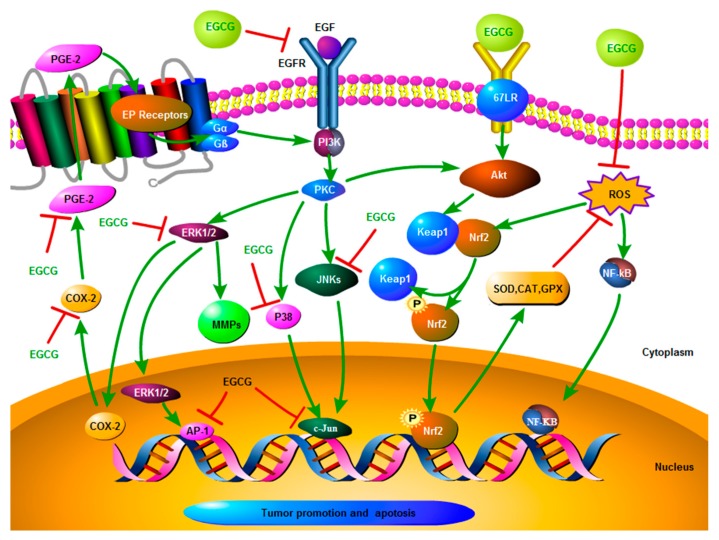
Possible mechanisms of EGCG on EC based on published literatures. **Note**: A red symbol” T” means “Inhibition”; a green arrow means “Activation”. Abbreviation: PKC: protein kinase C; PI3K, phosphatidylinositol 3-kinase; Akt, protein kinase B; SOD, Superoxide dismutase; CAT, catalase; GPX, glutathione peroxidases; EP, E-series of Prostaglandin; G α/β, G protein α/β; Nrf2, nuclear related factor erythroid-derived 2; Keap1, Kelch like ECH associated protein 1; NF-kB, Nuclear factor kB.

**Table 1 molecules-24-00954-t001:** Inclusion and exclusion criteria for literature search.

Search Strategy	Details
Inclusion criteria	(1) The studies were performed on humans, animals, cells. (2) It focuses on the relationship of tea or polyphenol, or EGCG, catechins on EC. However, if one literature was not related with tea and EC but it is necessary for background information, we also cited it in the paper. (3) Full text is available. Reviews and articles are all accepted. (4) It is published in English. (5) If it was a case control or cohort study, and the NOS score ≥5.
Exclusion criteria	(1) Important data was not available in the article. (2) It has nothing to do with tea on EC and is not necessary for background. (3) Full text is not available. (4) It is not published in English. (5) It was a case control or cohort study, but the NOS score <5.
Time filter	None (from inception)
Databases	Pubmed/Medline, Embase and Web of science

**Table 2 molecules-24-00954-t002:** Epidemiological studies of green tea drinking on EC risk.

Author [Reference] Location Year	Study Type Number of Subjects/Participants	Green Tea Drinking: Frequency or Amount	Risk Estimate of RR (95% CI)	Comments
Zheng et al. [32] 2012	Meta-analysis 8 case control 2 cohort studies 33,731 participants 3557 cases	**Males and Females**		No significant association between green tea consumption and EC risk, but an evidence of protective effect was observed among female.
		Non-tea drinker	1.00	
		Tea drinker	0.86 (0.7–1.03)	
		**Males**		
		Non-tea drinker	1.00	
		Tea drinker	1.04 (0.49–1.59)	
		**Females**		
		Non-tea drinker	1.00	
		Tea drinker	0.43 (0.21–0.66)	
Nechuta et al. [28] China 2012	Prospective cohort study 69,310 participants	Non-tea drinker	1.00	Adjusted for age, marital status, education, occupation, BMI, exercise, fruit and vegetable intake, meat intake, diabetes, and family history of digestive system cancer.
		Tea drinker:		
		**Amount**		
		<100 g/month	0.87 (0.55–1.37)	
		100–150 g/month	0.74 (0.47–1.17)	
		≥150 g/month	0.76 (0.48–1.19)	
		Non-tea drinker	1.00	
		Tea drinker:		
		**Duration**		
		<10 years	0.85 (0.55–1.32)	
		10–19 years	0.77 (0.46–1.28)	
		≥20 years	0.74 (0.49, 1.14)	
		**Overall**		
		Non-tea drinker	1.00	
		Green tea drinker	0.77 (0.57–1.03)	
Zheng et al. [30] 2013	Meta-analysis 14 case control 2 cohort studies 487,894 controls 7376 cases 8,874,734 participants	**Overall**		Green tea consumption was slightly inversely associated with EC risk, and it was more evident in Chinese population. No protective effect was found for black tea consumption.
		Non-tea drinker	1.00	
		Green tea drinker	0.77 (0.57–1.04)	
		**China**		
		Non-tea drinker	1.00	
		Green tea drinker	0.64 (0.44–0.95)	
Sang et al. [31] 2013	Meta-analysis 10 case control 2 cohort studies 487,894 controls 3821 cases	**Males and Females**		No significant association between green tea consumption and risk of EC. However, subgroup analysis showed a significant reduction (54%) in risk of EC in women with the highest green tea consumption compared with no/occasional drinkers.
		Non-tea drinker	1.00	
		Tea drinker	1.14 (0.97–1.35)	
		Moderate Drinker	0.94 (0.77–0.13)	
		Little-drinker	0.97 (0.77–1.22)	
		**Females**		
		Non-tea drinker	1.00	
		Tea drinker	0.46 (0.29–0.73)	
Oze et al. [39] Japan 2014	Hospital based case control study 961/2883	**Frequency**		Models included age, sex, coffee and green tea intake, cumulative smoking, alcohol consumption, fruit and vegetable intake, body mass index, occupation and frequency of rice intake.
		Less than 1cup/day	1.00	
		1 cup/day	1.20 (0.82–1.77)	
		2 cups/day	1.00 (0.65–1.65)	
		≥3 cups/day	1.31 (0.95–1.81)	
Zamora-Ros et al. [40] 9 European countries 2014	Prospective cohort study 442,143 participants	Non-tea drinker	1.00	Adjusted for center, sex, age, educational level, smoking status and intensity, physical activity, energy intake, daily consumption of fruit, vegetables, red and processed meat and coffee and tea mutually.
		Green tea drinker		
		**Amount**		
		<178.6 mg/d	0.85 (0.60–1.20)	
		≥178.6 mg/d	0.74 (0.51–1.08)	
Das et al. [34] India 2015	Hospital based case control study 39/41	Tea drinker		Drinking tea ≥ 3 cups/day, the occurrence rate of ESCC increased.
		**cups/day**		
		2	*p* = 0.0003	
		3		
		4		
Tai et al. [41] China 2017	Population-based case-control study 167/167	Tea temperature:		Age, sex, education level, body mass index, smoking status, alcohol drinking, family history of cancer in first degree relatives, and daily intakes of vegetables and fruits
		Low or mild (<60 °C)	1.0	
		High (≥60 °C)	2.23 (1.45–2.90)	
Yang et al. [42] China 2018	Population based case control study 1355/1962	Never tea drinking	1.00	Adjusted for age, marital status, education, occupation, family wealth score, body mass index 10 years ago, sum of missing and filled teeth, number of tooth brushing per day, smoking pack-years, alcohol consumption intensity and family history of EC among first-degree relatives.
		Hot tea drinking	2.15 (1.52–3.05)	
Yu et al. [43] China 2018	Population based cohort study 456,155 participants	**Frequency ***		Adjusted for age, sex, education, marital status, household income, physical activity, intake of red meat, fresh fruits and vegetables and preserved vegetables, body mass index, family history of cancer, and tobacco smoking. *: Participants who consumed pure alcohol <15 g/day or didn’t drink alcohol everyday **: Participants who consumed pure alcohol >15 g/day All of the data was calculated with participants who consumed tea less than weekly and consumed <15 g/d of pure alcohol as the reference category.
		Less Than Weekly	1.00	
		Weekly	0.82 (0.57–1.18)	
		Daily		
		Warm	0.92 (0.66–1.30)	
		Hot	1.23 (0.96–1.59)	
		Burning hot	1.36 (1.00–1.86)	
		**Frequency ****		
		Less Than Weekly	1.90 (1.57–2.31)	
		Weekly	2.60 (1.79–3.76)	
		Daily		
		Warm	3.74 (2.86–4.90)	
		Hot	3.84 (3.06–4.83)	
		Burning hot	5.00 (3.64–6.88)	

**Table 3 molecules-24-00954-t003:** Anticancer effects of green tea and its bioactive constituents on EC in in vitro and in vivo studies.

Author/ Reference	Compound/ Doses	Cell Line	Animal Model		Observed Effects
Ye et al. [17]	EGCG 20–40 μM	TE-8 SKGT-4	NA	↓ ↑	Cell proliferation, Invasion, pERK1/2, c-Jun, COX-2 Caspase-3
Ye et al. [17]	EGCG 50 μg/kg/day	NA	Nude mouse xenograft	↓	Tumor growth, Ki67, pERK1/2, COX-2
Hou et al. [19]	EGCG 5–50 μM	KYSE150 OE-19	NA	↓	EGFR, pEGFR, HER-2/neu, pHER-2/neu, PDGFRβ and colony formation
Liu et al. [22]	EGCG 0–400 μM	Eca-109 Te-1	NA	↓ ↑	Proliferation, Cell cycle, VEGF Apoptosis, ROS, cleaved caspase-3
Liu et al. [22]	EGCG 10 mg/kg/day	NA	Nude mouse xenograft	↓ ↑	VEGF, Tumor growth Cleaved-caspase-3
Gao et al. [23]	EGCG 50 μM	Eca-109	NA	↑	Apoptosis, Caspase-3, Caspase-9, JNK, P38
Meng et al. [44]	EGCG 0–200 mg/L	Eca-109	NA	↓ ↑	Cell proliferation Apoptosis, p16 gene demethylation
Liu et al. [45]	EGCG 0–400 mg/L	Ec-9706 Eca-109	NA	↓ ↑	Cell proliferation, telomerase activity, mitochondrial membrane potential Apoptosis, Caspase-3
Liu et al. [46]	EGCG 0–400 mg/L	Ec-9706 Eca-109	NA	↓ ↑	Cell proliferation, Bcl-2 Apoptosis, Bax, Caspase-3
Li et al. [47]	EGCG4–10 mg/kg	NA	NMBA-induced F344 rat	↓	Cyclin D1, COX-2, PGE-2, tumor growth, EC incidence rate
Song et al. [48]	Polyphenon E 20–40 μg/mL	SEG-1 BIC-1 SKGT-4 BE-3	NA	↓ ↑	Cell proliferation, Cyclin D1, Apoptosis, dephosphorylation of Rb
Chen et al. [49]	Green tea 1–5 g	NA	NMBA-induced Wistar rat	↓	Tumor incidence, tumor growth, DNA methylation, urinary N-nitrosoproline (NPRO) excretion, incidences of general lesions and precancerous lesions
Fang et al. [50]	EGCG 5–50 μM	KYSE 510	NA	↓ ↑	DNMT, cell growth Reversal of hypermethylation and reactivation of RARβ, MGMT, p16^INK4a^ and hMLH1
Morse et al. [51]	EGCG 360 or 1200 ppm	NA	NMBA-induced F344 rat	↓	Tumor incidence, tumor multiplicity
Wang et al. [52]	0.6% or 0.9% green tea extract	NA	NMBA- induced rat model	↓	Tumor incidence, tumor multiplicity, tumor growth

Note: NA, No available; ↓ Inhibit; ↑ Promote. Abbreviation: pERK, phosphorylated extracellular regulated protein kinases; COX-2, Cyclooxygenase-2; EGFR, epidermal growth factor receptor; pEGFR, phosphorylated epidermal growth factor receptor; HER-2, human epidermal growth factor receptor-2; PDGFRβ, platelet—derived growth factor receptor β; VEGF, vascular endothelial growth factor; ROS, reactive oxygen species; JNK, c-Jun N-terminal kinase; BCL-2, B-cell lymphoma-2; Prostaglandin E2, PGE-2; MGMT, O6-methylguanine methyltransferase; hMLH1, human mutL homologue 1; RARβ, retinoic acid receptor; NMBA, N-Nitrosomethylbenzylamine; DNMT, DNA methyltransferases.

**Table 4 molecules-24-00954-t004:** Pharmaceutical synergistic effects of EGCG with other treatments on EC.

Reference	Ingredient	Drug	Cell Line	Cytotoxic Action
Ye et al. [17] 2012	EGCG 20–40 μM or 50 μg/kg/day	Curcumin, lovastatin, or curcumin and lovastatin	SKGT-4 TE-8	Suppressing tumor cell viability and invasion; inhibiting xenograft tumor growth in nude mouse through downregulating the expression of p-ERK1/2, c-Jun and COX-2; upregulating caspase 3 expression.
Gao et al. [19] 2013	EGCG 50 μM	Vitamin C	Eca-109	Vitamin C could enhance the therapeutic properties of EGCG, activate caspase-3/9, induce apoptosis and regulate MAPK pathways.
Hou et al. [23]	EGCG 5–50 μM	SOD	KYSE150	EGCG was stabilized by SOD, and the growth inhibitory effect of EGCG on EC cell was potentiated by downregulating the activity of EGFR or HER-2/neu.
Liu et al. [46] 2017	EGCG 0–400 mg/L	ADM	Eca-109	EGCG promoted the rate of apoptosis and reversal of multidrug resistance induced by ADM through reducing the ABCG2 expression of Eca109/ABCG2 cells.

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
