# Peer review of "Inhibitory Effects of (−)-Epigallocatechin-3-gallate on Esophageal Cancer"

_molecules, 2019, doi:10.3390/molecules24050954_

Round 1
Reviewer 1 Report
Authors should demonstrate why this paper need to be published even though there are recent review papers regarding to EGCG and green tea on cancer treatment.
Author Response
Dear Editors and Reviewers:
We are grateful to your letter and the reviewers’ comments concerning our manuscript entitled “Inhibitory Effects of (-)-Epigallocatechin-3-gallate on Esophageal Cancer” (molecules-451026). Those comments are all valuable and very helpful for revising and improving our paper, as well as important guiding significance to our review. We have studied comments carefully and have revised the paper and provided point-by-point responses to their comments as following.
The manuscripts we submitted include a “revised version” and a “revised clean version” (final version). The revised parts were marked in red ink on the revised version, which was submitted for your consideration for publication in your great journal.
Thank you very much for your great helps and supports.
Responds to the Reviewers’ Comments
Reviewer #1
Comments and Suggestions for Authors
Authors should demonstrate why this paper need to be published even though there are recent review papers regarding to EGCG and green tea on cancer treatment.
Responds: Thank you for your comments. The responds are as follows:
Although there have been several review papers regarding to EGCG or green tea on cancer risk, only one review paper focusing on the effects of tea on EC prevention and treatment was published in 2011 [20] based on our literature searching from databases of Embase, PubMed/ Medline and Web of Science up to early February, 2019. However, many studies have documented the preventive effects of tea or EGCG on EC since 2012 [21-23], and also the underlying molecular mechanisms of EGCG on EC remain to be elucidated. Reviewing the recent advances in this topic will be help us to further understanding the relationship between tea drinking and EC risk.
In this review, epidemiologic investigations, studies using cell and animal models regarding to molecular and signaling pathways as well as pharmaceutical synergistic effects associated with green tea or EGCG on EC were summarized. We expect it will facilitate to develop natural products including tea extracts for EC prevention and treatment. We have added the background information in the introduction. (See P2L48-61 in the revised version)
Reference
[20 Yuan, J. M. Green tea and prevention of esophageal and lung cancers. Mol Nutr Food Res 2011, 55, 886-904.
[21]Xie, Y.; Huang, S.; Su, Y. Dietary Flavonols Intake and Risk of Esophageal and Gastric Cancer: A Meta-Analysis of Epidemiological Studies. Nutrients 2016, 8, 91.
[22]Liu, L.; Hou, L.; Gu, S.; Zuo, X.; Meng, D.; Luo, M.; Zhang, X.; Huang, S.; Zhao, X. Molecular mechanism of epigallocatechin-3-gallate in human esophageal squamous cell carcinoma in vitro and in vivo. Oncol Rep 2015, 33, 297-303.
[23]Gao, Y.; Li, W.; Jia, L.; Li, B.; Chen, Y. C.; Tu, Y. Enhancement of (-)-epigallocatechin-3-gallate and theaflavin-3-3'-digallate induced apoptosis by ascorbic acid in human lung adenocarcinoma SPC-A-1 cells and esophageal carcinoma Eca-109 cells via MAPK pathways. Biochem Biophys Res Commun 2013, 438, 370-374.
Reviewer 2 Report
I believe the readers could benefit from reading this manuscript. However, there are some issues that need to be addressed to enhance the clarity and comprehensiveness of this review article. Please see further details below.
Major points:
1. Please describe the literature review methods used (for example, how were the articles included in this manuscript retrieved? Was there a certain search strategy used? What were your inclusion and exclusion criteria?
2. Please correct grammatical and spelling errors. Also, multiple sentences were written in a quite awkward and unnatural way.
3. Table 1. Please describe how the studies listed in Table 1 were selected among other epidemiological studies of green tea drinking on EC risk out there. What were your criteria?
4. Table 1 is quite difficult to read, because Risk estimate of RR of different categories was presented without any dividing lines or space in-between. Also, in Yu et al.’s study (2018), “warm”, “hot”, and “burning hot” are all daily tea drinkers according to the original article, but it is not described in the table.
5. There are some other concerns about the details presented in Table 1.
(A) while the countries that the data were collected were presented in part of the studies listed.
(B) For study participants, the number presented is the total participants for each study. However, in some cases, this could be somewhat misleading if the study is not exclusively included subjects related to tea drinking and esophageal cancer. For example, Ren et al.’s study (2010), 481,563 participants include other cancer types than esophageal cancer, and also included other types of beverages than teas. For Yu et al.’s study (2018), the hazard ratios presented were from the part of the participants not from all 456,155 participants (the subjects who had less than daily alcohol consumption or < 15 g/d of pure alcohol).
6. P11L255-257: “A prospective phase II clinical trial in China confirmed that oral administration of EGCG was an effective therapy method for acute respiratory infection”: this study was about the effect of EGCG for radiation-induced esophagitis, not for acute respiratory infection
Other points:
1. P1L31: great progresses have à great progress has
2. P1L44: EGCG: please spell out when it first appears in the body of the manuscript.
3. P2L49: dinking --> drinking
4. P2L71: Gao et al --> Gao et al.
5. P7L92: EGCG were --> EGCG was
6. P7L93: Synergistic effect --> A synergistic effect
7. P7L99: important fectors --> important factors
8. P7L107: involing --> involving
9. P7L110: barrett’s --> Barrett’s
10. P8L116: lovastin --> lovastatin
11. P8L119: a key regulating factors --> a key regulating factor
12. P8L133: The inhibitory efficacy of EGCG against angiogenesis have been reported --> The inhibitory efficacy of EGCG against angiogenesis has been reported
13. P8L147: please spell out DNMT when it first appears
14. P8L147: CPG --> CpG
15. P8L153: Fang et al --> Fang et al.
16. P9L178: There was animal study --> There was an animal study
17. P9L189: NO2 --> NO2
18. P9L195: I was found --> It was found
19. P10L220: Wang et al --> Wang et al.
20. P10L224: Li et al --> Li et al.
21. P11L279: please spell out PAH
22. P11L285: a lot catechins --> a lot of catechins
23. P12L294: plasm --> plasma
24. P12L312: EGCG have --> EGCG has
25. P12L317: a member cancer cells --> a member of cancer cells
26. P12L334-335: a promising targets --> a promising target
Author Response
Dear Editors and Reviewers:
We are grateful to your letter and the reviewers’ comments concerning our manuscript entitled “Inhibitory Effects of (-)-Epigallocatechin-3-gallate on Esophageal Cancer” (molecules-451026). Those comments are all valuable and very helpful for revising and improving our paper, as well as important guiding significance to our review. We have studied comments carefully and have revised the paper and provided point-by-point responses to their comments as following.
The manuscripts we submitted include a “revised version” and a “revised clean version” (final version). The revised parts were marked in red ink on the revised version, which was submitted for your consideration for publication in your great journal.
Thank you very much for your great helps and supports.
Responds to the Reviewers’ Comments
Reviewer #2
Comments and Suggestions for Authors
I believe the readers could benefit from reading this manuscript. However, there are some issues that need to be addressed to enhance the clarity and comprehensiveness of this review article. Please see further details below.
Major points:
1. Please describe the literature review methods used (for example, how were the articles included in this manuscript retrieved? Was there a certain search strategy used? What were your inclusion and exclusion criteria?
Responds: Thank you for your question. The responds are as follows:
Published literatures regarding to tea or EGCG on EC were searched from databases of Embase, PubMed/ Medline and Web of Science up to Dec 13th, 2018. The searching terms used were: (1) “tea” or “green tea” “EGCG” or “polyphenols” or “catechins”; (2) “esophageal carcinoma “or “esophageal cancer “or “esophageal neoplasms”; (3) these searched key words were combined using “and” without restrictions. We retrieved all the searched papers and removed the duplicate references by Endnote. Then, we read the titles and abstracts of the remaining papers to exclude those which were not related to the research topic. When we could not decide whether a paper should be included or not based on the title and abstract, we read the full text of the article. The searched papers were screened according to the inclusion and exclusion criteria in Table 1. The screened papers were then classified into epidemiological studies, animal and cell studies, clinical trials and inconsistent results, which will be cited in the corresponding subtitles.
Table 1. Inclusion and exclusion criteria for literature search
Search Strategy | Details |
Inclusion criteria | ①. Studies performed on humans, animals, cells. ②. It focuses on the relationship of tea or polyphenol, or EGCG, catechins on EC. However, if one literature was not related with tea and EC but it is necessary for background information, we also cited it in the paper. ③. Full text is available. Reviews and articles are all accepted. ④. It is published in English. |
Exclusion criteria | ①. Important data was not available in the article. ②. It has nothing to do with tea on EC and is not necessary for background. ③. Full text is not available. ④. It is not published in English. |
Time filter | None (from inception) |
Databases | Pubmed/Medline, Embase and Web of science |
The revision was included in the Section 2. Literature search strategy. (See P2L62-75 in the revised version)
2. Please correct grammatical and spelling errors. Also, multiple sentences were written in a quite awkward and unnatural way.
Response to the comments: We rechecked the context and the sentences were polished by a native English speaker. The revised parts were all marked in red in the revised version.
3. Table 1. Please describe how the studies listed in Table 1 were selected among other epidemiological studies of green tea drinking on EC risk out there. What were your criteria?
Responds: Thank you for your question. The included studies were based on the following two steps.
For the literatures under the subtitle “epidemiological studies”, each was critically appraised by a quality scale, namely, the Newcastle-Ottawa Scale (NOS) [S1], which consists of three variables of quality as follows: selection (4 points), comparability (2 points), and outcome (3 points), for a total score of 9 points (9 representing the highest quality). The studies to be included in Table 2 should meet the following criteria: ①. It was a case-control or cohort study. ②. It tested the association between EC risk and tea drinking; the odds ratios (ORs) or adjusted OR values and relevant corresponding 95% confidence intervals (CIs) were reported. If the OR value is not available, P value was also accepted. ③. The study was about tea individually or tea was definitely included in the study. ④. It was published in English. ⑤. The NOS score ≥ 5. However, the literatures regarding to the epidemiological studies may be cited in the other part of the paper for background information.
The above information was included in Section 2. Literature search strategy.
(See P3L76-85 in the revised version)
4. Table 1 is quite difficult to read, because Risk estimate of RR of different categories was presented without any dividing lines or space in-between. Also, in Yu et al.’s study (2018), “warm”, “hot”, and “burning hot” are all daily tea drinkers according to the original article, but it is not described in the table.
Responds: Thanks for your advice. We have added short dividing lines to separate different categories in Table 1 as your suggestion, and the subtitle in every category has been bolded. Also, we have checked Yu et al.’s study [43] carefully and added “daily” in the table. We are very sorry for our carelessness. (See P6 in the revised version)
References
[43] Yu, C.; Tang, H.; Guo, Y.; Bian, Z.; Yang, L.; Chen, Y.; Tang, A.; Zhou, X.; Yang, X.; Chen, J.; Chen, Z.; Lv, J.; Li, L. hot tea consumption and its interactions with alcohol and tobacco use on the risk for esophageal cancer: A population-based cohort study. Ann Intern Med 2018, 168, 489-497.
5. There are some other concerns about the details presented in Table 1.
(A) while the countries that the data were collected were presented in part of the studies listed.
Responds: Thank you for your comment. For your question, we have the following explanation. We have presented the country of each individual study, including each case control study and cohort study. But for the 3 meta- analysis [30-32], as it is a systematic review of studies that have been published in many different countries, we don’t list any one country of them. (See Table 2 P4-P5 in the revised version)
References
[30]Zheng, J. S.; Yang, J.; Fu, Y. Q.; Huang, T.; Huang, Y. J.; Li, D. Effects of green tea, black tea, and coffee consumption on the risk of esophageal cancer: a systematic review and meta-analysis of observational studies. Nutr Cancer 2013, 65, 1-16.
[31]Sang, L. X.; Chang, B.; Li, X. H.; Jiang, M. Green tea consumption and risk of esophageal cancer: a meta-analysis of published epidemiological studies. Nutr Cancer 2013, 65, 802-812.
[32]Zheng, P.; Zheng, H. M.; Deng, X. M.; Zhang, Y. D. Green tea consumption and risk of esophageal cancer: a meta-analysis of epidemiologic studies. BMC Gastroenterol 2012, 12,165.
(B) For study participants, the number presented is the total participants for each study. However, in some cases, this could be somewhat misleading if the study is not exclusively included subjects related to tea drinking and esophageal cancer. For example, Ren et al.’s study (2010), 481,563 participants include other cancer types than esophageal cancer, and also included other types of beverages than teas. For Yu et al.’s study (2018), the hazard ratios presented were from the part of the participants not from all 456,155 participants (the subjects who had less than daily alcohol consumption or < 15 g/d of pure alcohol).
Responds: We are very sorry for the mistake. For your comment, the explanations and corrections are as follows:
1)About Ren et al.’s study [29], although the total participants including other cancer types except esophageal cancer and the beverages including other types except tea, it doesn’t affect that the authors calculate the ORs by using unconditional logistic regression analysis and adjusting for other confounders.
However, based on the aforementioned selection criteria of the epidemiological study, we have removed Ren et al.’s study [29] and Chen et al.’ study [S2] from the table, because they have been included in the meta-analysis [30]. (SEE P5 in the revised version)
2) In regard to Yu et al.’s study (2018) [43], we have checked the paper again carefully and added ”Daily” and the part of “Pure alcohol consumption >15 g/day” in Table 2.
(SEE P6 in the revised version)
References
[29]Ren, J. S.; Freedman, N. D.; Kamangar, F.; Dawsey, S. M.; Hollenbeck, A. R.; Schatzkin, A.; Abnet, C. C. Tea, coffee, carbonated soft drinks and upper gastrointestinal tract cancer risk in a large United States prospective cohort study. Eur J Cancer 2010, 46, 1873-1881.
[30] Zheng, J. S.; Yang, J.; Fu, Y. Q.; Huang, T.; Huang, Y. J.; Li, D. Effects of green tea, black tea, and coffee consumption on the risk of esophageal cancer: a systematic review and meta-analysis of observational studies. Nutr Cancer 2013, 65, 1-16.
[43] Yu, C.; Tang, H.; Guo, Y.; Bian, Z.; Yang, L.; Chen, Y.; Tang, A.; Zhou, X.; Yang, X.; Chen, J.; Chen, Z.; Lv, J.; Li, L. hot tea consumption and its interactions with alcohol and tobacco use on the risk for esophageal cancer: A population-based cohort study. Ann Intern Med 2018, 168, 489-497.
[S2]Chen, Z.; Chen, Q.; Xia, H.; Lin, J. Green tea drinking habits and esophageal cancer in southern China: a case-control study. Asian Pac J Cancer Prev 2011, 12, 229-233.
6. P11L255-257: “A prospective phase II clinical trial in China confirmed that oral administration of EGCG was an effective therapy method for acute respiratory infection”: this study was about the effect of EGCG for radiation-induced esophagitis, not for acute respiratory infection
Responds: Thank you for pointing our mistake. We have read the literature carefully again and corrected the mistake. P14L296
7. Other points:
1. P1L31: great progresses have à great progress has P1L30-31
2. P1L44: EGCG: please spell out when it first appears in the body of the manuscript. P1L44-45
3. P2L49: dinking --> drinking P2L56
4. P2L71: Gao et al --> Gao et al. P3L103
5. P7L92: EGCG were --> EGCG was P7L123
6. P7L93: Synergistic effect --> A synergistic effect P7L124
7. P7L99: important fectors --> important factors P10L130
8. P7L107: involing --> involving P10L139
9. P7L110: barrett’s --> Barrett’s P10L141
10. P8L116: lovastin --> lovastatin P10L146
11. P8L119: a key regulating factors --> a key regulating factor P10L149
12. P8L133: The inhibitory efficacy of EGCG against angiogenesis have been reported --> The inhibitory efficacy of EGCG against angiogenesis has been reported P10L162
13. P8L147: please spell out DNMT when it first appears P10L175
14. P8L147: CPG --> CpG P11L176
15. P8L153: Fang et al --> Fang et al. P11L182
16. P9L178: There was animal study --> There was an animal study P11L210
17. P9L189: NO2 --> NO2 P12L230
18. P9L195: I was found --> It was found P12L236
19. P10L220: Wang et al --> Wang et al. P13L260
20. P10L224: Li et al --> Li et al. P13L264
21. P11L279: please spell out PAH (polycyclic aromatic hydrocarbons) P15L318
22. P11L285: a lot catechins --> most of catechins P15L325
23. P12L294: plasm --> plasma P15L334
24. P12L312: EGCG have --> EGCG had P15L352
25. P12L317: a member cancer cells --> a member of cancer cells P15L356
26. P12L334-335: a promising targets --> a promising target P16L374
Responds: Thank you for your careful review of the manuscript. We are very sorry for our incorrect writing and have corrected them according to your suggestion. All changed words are marked in red in the revised version.
Reviewer 3 Report
The manuscript presents a review of few papers showing the effect of green tea drinking on esophageal cancer. A review is presented in nice form, however, I have the impression that the study does not consider a wide spectrum of research on other effects of tea catechins on human health, also on cancer.
- Recently, Furushima et al. (Molecules 2018 23(7)) has published a review on the effect of tea catechins on influenza infections. Together with the study presented in the manuscript suggests that the effect of cancer inhibition is indirectly linked with cancer progression as catechins seems to have good influence on human health. The authors should linked and comment their studies with Furishima work. In my opinion, saying that EGCG inhibits esophaegeal cancer is to speculative, especially that the authors alone present also controversial results.
- Some idea of the possible EGCG mechanism in cancer cells is presented in the paper by Suganuma et al. Molecules. 2016 Nov 18;21(11). pii: E1566. It should be included in the discussion.
- It take a while to understand the scheme presented in Figure 1. The plot is not clear. In conjunction with above remark, I would include more detailed description of potential, most probable signalling pathways, underlying these processes and molecules that are specific for cancer progression.
- What is the optimal dose of EGCG ? How it relates to the number of green tea cups ?
Author Response
Dear Editors and Reviewers:
We are grateful to your letter and the reviewers’ comments concerning our manuscript entitled “Inhibitory Effects of (-)-Epigallocatechin-3-gallate on Esophageal Cancer” (molecules-451026). Those comments are all valuable and very helpful for revising and improving our paper, as well as important guiding significance to our review. We have studied comments carefully and have revised the paper and provided point-by-point responses to their comments as following.
The manuscripts we submitted include a “revised version” and a “revised clean version” (final version). The revised parts were marked in red ink on the revised version, which was submitted for your consideration for publication in your great journal.
Thank you very much for your great helps and supports.
Responds to the Reviewers’ Comments
Reviewer #3
Comments and Suggestions for Authors
The manuscript presents a review of few papers showing the effect of green tea drinking on esophageal cancer. A review is presented in nice form, however, I have the impression that the study does not consider a wide spectrum of research on other effects of tea catechins on human health, also on cancer.
1.Recently, Furushima et al. (Molecules 2018 23(7)) has published a review on the effect of tea catechins on influenza infections. Together with the study presented in the manuscript suggests that the effect of cancer inhibition is indirectly linked with cancer progression as catechins seems to have good influence on human health. The authors should linked and comment their studies with Furishima work. In my opinion, saying that EGCG inhibits esophageal cancer is too speculative, especially that the authors alone present also controversial results.
Responds: Thank you for your comment. The reply is as follows:
1) Response to the comments “the study does not consider a wide spectrum of research on other effects of tea catechins on human health, also on cancer”.
Responds: We have carefully read the literature mentioned and searched the recent advances on other health effects of EGCG on human body. Then, we added the antiviral infection effects and 3 other latest references in the introduction [9-12] (These were added in P1-P2 L46-48 in the revised version).
Furthermore, it was reported that tea catechins inhibited influenza viral adsorption and suppressed replication and neuraminidase activity [10]. Tea catechins enhance immunity against viral infection, suggesting that the cancer inhibition effect of tea catechins might also be indirectly linked with cancer progression as catechins seems to have good influence on human health. (These were added in P16L395-399 in the revised version).
2) Response to the comments “In my opinion, saying that EGCG inhibits esophaegeal cancer is too speculative, especially that the authors alone present also controversial results.”
Responds: We have added “in laboratory studies” in the conclusion, and it might be more reasonable. The revised conclusion is” Overall, EGCG shows an inhibitory effect on EC in laboratory studies.” (P17L410-411 in the revised version)
3) Responds to the comments” Together with the study presented in the manuscript suggests that the effect of cancer inhibition is indirectly linked with cancer progression as catechins seems to have good influence on human health”.
Responds: Thanks. What you have pointed out is a very noticeable topic.
As we know, Green tea is a rich source of pharmacologically active molecules, such as EGCG, a natural antioxidant agent, which has been implicated to provide diverse health benefits to human body.
On one hand, EGCG plays an important role in cancer prevention indirectly through the activity of its anti-oxidation, pro-oxidation and the regulation of related signaling pathways, such as MAPK signaling pathway [23, 74]. (P11L196-199 in the revised version).
On the other hand, EGCG has been confirmed that it also could exert the anticancer effects by directly binding to several proteins, such as the trans-membrane receptor 67 kDa laminin receptor (67LR), anti-apoptotic protein BCL-2, and thus affecting cancer cell proliferation and apoptosis. In addition, EGCG was also found to be interacting directly with Pin1, TGFR-II, and metalloproteinases (MMPs) [73].We have included the related paper in the review. (P11L193-195 in the revised version)
References
[9]Oliviero, F.; Scanu, A.; Zamudio-Cuevas, Y.; Punzi, L.; Spinella, P. Anti-inflammatory effects of polyphenols in arthritis. J Sci Food Agr 2018, 98, 1653-1659.
[10]Furushima, D.; Ide, K.; Yamada, H. Effect of Tea Catechins on Influenza Infection and the Common Cold with a Focus on Epidemiological/Clinical Studies. Molecules 2018, 23.
[11]Polito, C.; Cai, Z. Y.; Shi, Y. L.; Li, X. M.; Yang, R.; Shi, M.; Li, Q. S.; Ma, S. C.; Xiang, L. P.; Wang, K. R.; Ye, J. H.; Lu, J. L.; Zheng, X. Q.; Liang, Y. R. Association of Tea Consumption with Risk of Alzheimer’s Disease and Anti-Beta-Amyloid Effects of Tea. Nutrients 2018, 10, 655.
[12]Eng, Q. Y.; Thanikachalam, P. V.; Ramamurthy, S. Molecular understanding of Epigallocatechin gallate (EGCG) in cardiovascular and metabolic diseases. J Ethnopharmacol 2018, 210, 296-310.
[23]Gao, Y.; Li, W.; Jia, L.; Li, B.; Chen, Y. C.; Tu, Y. Enhancement of (-)-epigallocatechin-3-gallate and theaflavin-3-3'-digallate induced apoptosis by ascorbic acid in human lung adenocarcinoma SPC-A-1 cells and esophageal carcinoma Eca-109 cells via MAPK pathways. Biochem Biophys Res Commun 2013, 438, 370-374.
[73]Negri, A.; Naponelli, V.; Rizzi, F.; Bettuzzi, S. Molecular Targets of Epigallocatechin-Gallate (EGCG): A Special Focus on Signal Transduction and Cancer. Nutrients 2018, 10.
[74] Wagner, E. F.; Nebreda, A. R. Signal integration by JNK and p38 MAPK pathways in cancer development. Nat Rev Cancer 2009, 9, 537-549.
2.Some idea of the possible EGCG mechanism in cancer cells is presented in the paper by Suganuma et al. Molecules. 2016 Nov 18;21(11). pii: E1566. It should be included in the discussion.
Responds: Thank you for your suggestion. It is really a good and systematic review on possible EGCG mechanism in cancer cells. We have added the reference [117] in the review and made comments about it in the discussion. (P16L379-385 in the revised version)
References
[117]Suganuma, M.; Takahashi, A.; Watanabe, T.; Iida, K.; Matsuzaki, T.; Yoshikawa, H. Y.; Fujiki, H. Biophysical Approach to Mechanisms of Cancer Prevention and Treatment with Green Tea Catechins. Molecules 2016, 21.
3.It take a while to understand the scheme presented in Figure 1. The plot is not clear. In conjunction with above remark, I would include more detailed description of potential, most probable signaling pathways, underlying these processes and molecules that are specific for cancer progression.
Responds: Sorry for the confusion. In order to make it clear, we have added a table (Table 3) (P8-P9 in the revised version) to show the role of tea ingredients on esophageal cancer.
Also, a new figure (Figure 1) (P12 in the revised version) was drawn to show the relevant signaling pathways between EC and EGCG.
4. What is the optimal dose of EGCG? How it relates to the number of green tea cups?
Responds: Thank you for your question. We have checked the related references again and added the used doses of EGCG of each study in Table 3 (P8-P9 in the revised version) and Table 4 (P13-P14 in the revised version).
To give an optimal oral administration dose of EGCG, we checked the published paper recently. Khan et al. [118] have reported that drinking one cup of green tea contains up to 200 mg of EGCG daily, which has been shown to have chemo-preventive or chemotherapeutic effects against several types of cancers. Proper drinking of green tea is three to five cups per day, which is equivalent to a minimum of 1000 mg of EGCG per day.
Suganuma et al. [117] also offer a proposal that drinking over 10 cups of green tea per day (equivalent to 25 -30 g green tea) can prevent cancer recurrence in various organs in humans.
Based on the content of EGCG in dried green tea (about 80 mg/g) [120] and the results of epidemiological and clinical trials, and in consideration of the extraction rate of the tea catechins, we recommend that we can have 5-7 cups of green tea at a lower temperature one day, as it may be an effective method for the prevention of esophageal cancer.
In addition, we have added the related reference and a suggestion in the discussion. (P16L386-399 in the revised version)
References
[117] Suganuma, M.; Takahashi, A.; Watanabe, T.; Iida, K.; Matsuzaki, T.; Yoshikawa, H. Y.; Fujiki, H. Biophysical Approach to Mechanisms of Cancer Prevention and Treatment with Green Tea Catechins. Molecules 2016, 21.
[118] Khan, N.; Mukhtar, H. Tea Polyphenols in Promotion of Human Health. Nutrients 2019, 11.
[120] Liang, Y.R.; Lu, J.L.; Shang, S.L. Effects of gibberellins on chemical composition and quality of tea (Camellia sinensis L). J Sci Food Agric 1996, 72, 411-414.
Round 2
Reviewer 1 Report
Why this manuscript focuses especially on ES?, What benefit this artivle can give to readers?
What is the difference in the mechanism of EGCG's action on esophageal cancer and other cancers?
Reviewer 2 Report
Most of my concerns from my previous review have been appropriately addressed, and the revised manuscript is significantly improved from the previous version.
Reviewer 3 Report
The authors explained and clarified all issues raised by me in a satisfactory way, therefore, I am recommending the manuscript for publication.